# Depressive Symptoms as a Mediator between Excessive Daytime Sleepiness and Suicidal Ideation among Chinese College Students

**DOI:** 10.3390/ijerph192316334

**Published:** 2022-12-06

**Authors:** Yao-Kun Yu, Zhi-Ying Yao, Yan-Xin Wei, Chang-Gui Kou, Bin Yao, Wen-Jun Sun, Su-Yun Li, Kenneth Fung, Cun-Xian Jia

**Affiliations:** 1Department of Epidemiology, School of Public Health, Cheeloo College of Medicine, Shandong University & Shandong University Center for Suicide Prevention Research, 44 West Wenhua Road, Jinan 250012, China; 2Department of Epidemiology and Biostatistics, School of Public Health, Jilin University, Changchun 130021, China; 3Student Mental Health Education and Counseling Center, Xi’an Jiaotong University, Xi’an 710049, China; 4Student Work Office, College of Textile and Clothing, Qingdao University, Qingdao 266071, China; 5Department of Epidemiology and Health Statistics, School of Public Health, Qingdao University, Qingdao 266071, China; 6Department of Psychiatry, University of Toronto, Toronto Western Hospital, 399 Bathurst St. 9 EW, Toronto, ON M5T2S8, Canada

**Keywords:** excessive daytime sleepiness, depressive symptoms, suicidal ideation, college students, mediation analysis

## Abstract

The purpose of this study was to verify the mediating role of depressive symptoms between excessive daytime sleepiness and suicidal ideation in college students. Of the 6944 participants, 2609 (37.6%) were male and 4335 (62.4%) were female. College students with excessive daytime sleepiness (*p* < 0.001) and those with depressive symptoms (*p* < 0.001) were more likely to have suicidal ideation. Moreover, both excessive daytime sleepiness (β = 0.14, 95% CI: 1.01–1.32) and depressive symptoms (β = 1.47, 95% CI: 3.80–5.00) were associated with suicidal ideation. The effect size of the mediating role of depressive symptoms in excessive daytime sleepiness to suicidal ideation was 50.41% for the entire sample, 58.33% for males, and 42.98% for females. Depressive symptoms partially mediated the relationship between excessive daytime sleepiness and suicidal ideation. The timely assessment of depressive symptoms in college students with excessive daytime sleepiness, and intervention, may reduce their risk of suicidal ideation.

## 1. Introduction

Suicide is a worldwide health problem of concern [1,2]. Reliable studies have reported that the number of deaths by suicide exceeds 700,000 per year globally, which brings a serious economic and disease burden for countries around the world [3]. Currently, in the midst of the COVID-19 pandemic, which has adversely affected the mental health of both people with mental illness and healthy populations and is directly related to multiple suicide triggers, such as stress, unemployment, and alcohol use, the situation is extremely negative for suicide prevention [4]. Moreover, suicide is the fourth leading cause of death in the 15–29 age group and has become a public health issue that cannot be ignored in this age group [3,5]. According to the three-step theory of suicide, suicidal ideation (SI) is the antecedent step to suicide attempt (SA) [1]. Suicidal ideation is strongly predictive of suicide deaths [1]. A study in the US has documented that the prevalence of SI among college students is about 24% [6]. This is much higher than the global prevalence of SI among adults [6]. Therefore, the problem of college student suicide needs extra attention, especially the study of SI among college students.

College students are at a special stage of their lives and perceive changes in stress that are more likely to cause various mental health problems [7,8]. Depressive symptoms are one of the main problems of college students’ mental health [9]. Previous studies have demonstrated that SI may be the result of mild to moderate depression in college students [10]. The prevalence of depressive symptoms among Chinese college students continues to be high [11]. Currently, evidence suggests that depression may be caused by a variety of factors, including family factors [12,13], academic performance [14], and sleep disturbances [15,16]. Sleep disturbance, as an important risk factor for depression, can predict the onset and outcome of depression [15]. Numerous studies have demonstrated that sleep disturbances precede depression, and that depressed patients with sleep disturbances experience more severe symptoms and treatment difficulties [15,17].

Sleepiness is the tendency to doze off or fall asleep when you intend to stay awake. It is distinguished from tiredness or fatigue. Excessive daytime sleepiness (EDS) is a type of sleep disturbance [18]. Excessive daytime sleepiness may lead to a decrease in college students’ ability to study and work, an increase in the risk of accidents, and difficulties in interpersonal relationships, which will undoubtedly have a huge negative impact on college students’ physical and mental health [19]. Currently, multiple studies have demonstrated that EDS is associated with psychiatric disorders in college students [20,21,22]. A longitudinal study in the US demonstrated that EDS was associated with incident depression in both genders, with a stronger correlation in women aged 20–40 years [23]. Moreover, EDS is very common among college students. One study even observed a 55% prevalence of EDS among college students [21]. The mechanism of sleepiness may be due to hypometabolism in some areas of the brain [24]. Currently, during the COVID-19 pneumonia pandemic, sleep indicators among college students have deteriorated significantly, including sleep quality, sleep latency, sleep duration, sleep disturbance, and the increased use of sleep medication. Therefore, excessive daytime sleepiness among college students may be more severe [25].

Currently, there are few studies on EDS, depressive symptoms, and SI among Chinese college students. Although some studies have demonstrated that EDS, depressive symptoms, and SI are highly correlated [21,26], there is still a lack of research on the causes and mechanisms of the association between the three. Therefore, this study was conducted to investigate the possible mechanisms of EDS and depressive symptoms leading to SI by constructing a database of Chinese college students. In detail, we (1) explored the association between EDS, depressive symptoms, and SI among Chinese college students; and (2) examined whether depressive symptoms could mediate the association between EDS and SI among college students, as shown in Figure 1.

## 2. Materials and Methods

### 2.1. Participants and Procedures

From November 2020 to May 2021, we conducted a cross-sectional study of seven universities in China. Approximately 1000 university students from each university were recruited to participate in this survey through a cluster random sampling method. Participating students completed the survey through the online platform Questionnaire Star. Trained counselors distributed a link or QR code for the online survey to students’ WeChat groups, and informed participants that the survey was anonymous and voluntary. Before starting the survey, college students were required to read and agree to an informed consent form. A total of 9712 college students participated in the survey, and 6944 met the requirements of our study. Data were imported from Questionnaire Star into IBM SPSS Statistics 24.0 (Armonk, New York, NY, USA) and Stata MP 17.0 for analysis.

### 2.2. Measurements

#### 2.2.1. Excessive Daytime Sleepiness

The Epworth Sleepiness Scale [27] is a widely used scale to assess excessive daytime sleepiness with good reliability and validity [28]. The Epworth Sleepiness Scale is a four-point Likert scale ranging from 0 = not likely, 1 = very unlikely, 2 = moderately likely, and 3 = very likely. The scores of the eight items were summed to obtain a total score, which ranged from 0 to 24. The Cronbach’s α coefficient for this study was 0.73. In this study, a total score between 0 and 10 was defined as normal and a score greater than 10 was defined as excessive daytime sleepiness [27,29].

#### 2.2.2. Depression Symptoms

The depression subscale of the Depression Anxiety Stress Scale-21 (DASS-21) [30] was used to assess participants’ depressive symptoms. The scale has shown good reliability and validity in previous studies [31]. The subscale consists of 7 items with options of 0 = did not apply to me at all, 1 = applied to me to some degree or some of the time, 2 = applied to me to a considerable degree or a good part of the time, to 3 = applied to me very much or most of the time. The scores of the seven items were summed and multiplied by two to obtain a total score, which ranged from 0 to 42. College students scoring 0–9 were considered normal, and those scoring 10–42 exhibited depressive symptoms. The Cronbach’s α coefficient for this subscale in the sample was 0.87.

#### 2.2.3. Suicidal Ideation

Suicidal ideation was measured by a question, “How many times in the past year have you considered suicide?”. The responses were “never”, “rarely (1 time)”, “sometimes (2 times)”, “often (3–4 times)” and “frequently (5 times and more)”. In this study, students who responded ‘’once or more in the past year’’ were classified as having suicidal ideation and others were classified as not having suicidal ideation.

#### 2.2.4. Study Covariates

Factors included in the study as covariates were age, gender, body mass index (BMI, weight/height^2^, kg/m^2^), community (urban, rural), nationality (Han, others), grade, only child, major, smoking, drinking, academic performance (poor, fair, or good), economic status (poor, fair or good), being single or not, school activities (attended, never attended), daily game time, daily social media time, weekly sports time, and sleep quality (poor, general, or good). The above covariates were investigated through a self-administered questionnaire.

### 2.3. Statistical Analysis

Missing values for daily game time, daily media time, and weekly sports time in the study were processed using mean interpolation. None of the continuous variables in the study conformed to a normal distribution, so they were described by median (interquartile range, IQR), and categorical variables were described by frequency (proportion). Correlations between suicidal ideation and other variables were tested by chi-square test and Mann–Whitney U test. Correlations between gender, EDS, and depressive symptoms were verified by Spearman correlation analysis. Unconditional logistic regression was performed to test the relationship between EDS, depressive symptoms, and SI overall, in males and females. All the above analyses were completed by SPSS 24.0. The analysis of mediating effects was completed through Stata MP 17.0. As shown in Figure 1, the mediating effect model was established with EDS as the independent variable, depressive symptoms as the mediating variable, and SI as the dependent variable, controlling for covariates. The mediating effect model was described by total, direct, and indirect effects. The path of the direct effect was EDS—SI, the path of the indirect effect was EDS—depressive symptoms—SI, and the total effect was the sum of the two. All statistical analyses above were two-tailed tests, and *p* < 0.05 was considered statistically significant.

## 3. Results

### 3.1. Sample Characteristics

The characteristics of the sample college students are shown in Table 1. A total of 2609 (37.6%) males and 4335 (62.4%) females participated in this survey. Among the 6944 college students who participated in this survey, 1394 (20.1%) reported SI in the past year, and 5550 (79.9%) reported no SI. Females had a higher SI percentage than men (21.2% vs. 18.1%, *p* = 0.002). The chi-square test results showed that college students from urban areas (*p* < 0.001), being an only child (*p* = 0.009), medical major (*p* < 0.001), smoking (*p* < 0.001), drinking (*p* < 0.001), poor academic performance (*p* < 0.001), poor economic status (*p* < 0.001), never attending school activities (*p* < 0.001), and poor sleep quality (*p* < 0.001) were all associated factors for SI. The Mann–Whitney U test showed that daily game time, daily media time, and weekly sports time were all associated with SI (*p* < 0.001), and that longer time was associated with a lower risk of SI. In addition, there were no statistical differences between college students with SI and without SI in terms of age, BMI, nationality, and having a boy/girlfriend.

### 3.2. Excessive Daytime Sleepiness and Depressive Symptoms

The mean score of the Epworth Sleepiness Scale was 10.51 (SD = 4.18). A total of 3670 (52.9%) college students in this study exhibited EDS, and a higher percentage of college students with EDS had SI in the past year (62.0% vs. 50.6%, χ^2^ = 58.33, *p* < 0.001). The mean score on the Depression subscale of the DASS-21 was 6.03, SD was 7.13, and 1771 (25.5%) college students in this study had depressive symptoms. Depressive symptoms were highly positively correlated with SI (54.8% vs. 18.1%, χ^2^ = 788.21, *p* < 0.001).

### 3.3. Association between Gender, EDS and Depressive Symptoms

The Spearman correlation coefficient between EDS and depressive symptoms was 0.152 (*p* < 0.001). The Spearman correlation coefficients of EDS, depressive symptoms, and gender were 0.108 (*p* < 0.001) and -0.068 (*p* < 0.001), respectively.

### 3.4. Association between Excessive Daytime Sleepiness, Depressive Symptoms, and Suicidal Ideation

Spearman correlation analysis showed that gender was strongly correlated with all other covariates (*p* < 0.001). We will separate the study by gender. Table 2 demonstrates the associations between EDS, depressive symptoms, and SI in the total sample, male sample, and female sample. The multivariate logistic regression results showed that in the total sample, both EDS (β = 0.14, 95% CI: 1.01–1.32) and depressive symptoms (β = 1.47, 95% CI: 3.80–5.00) were correlates of SI. In the female sample, SI was also correlated with EDS (β = 0.19, 95% CI: 1.02–1.43) and depressive symptoms (β = 1.55, 95% CI: 3.95–5.60). However, only depressive symptoms (β = 1.32, 95% CI: 2.98–4.69) were associated with SI in males.

### 3.5. Mediating Effects of Depressive Symptoms

Table 3 presents the results of the mediated effects analysis with EDS as the independent variable, depressive symptoms as the mediating variable, and SI as the dependent variable. In the full sample, the total, direct, and indirect effects of EDS on SI were 0.29 (95% CI: 0.16–0.42), 0.14 (95% CI: 0.01–0.28), and 0.15 (95% CI: 0.11–0.18), respectively. The mediating effect size was 50.41%. The results of the analysis by gender showed that the total, direct, and indirect effects were 0.24 (95% CI: 0.03–046), 0.10 (95% CI: −0.12–0.32), and 0.14 (95% CI: 0.09–0.19) for the male sample and 0.33 (95% CI: 0.16–0.50), 0.19 (95% CI: 0.02–0.36), and 0.14 (95% CI: 0.09–0.19) for the female sample, respectively. However, the direct effect was not statistically significant in the male sample and the rest of the effects were statistically significant (*p* < 0.05). The indirect effect sizes for depressive symptoms in EDS to SI were 58.33% and 42.98% for males and females, respectively.

## 4. Discussion

As far as we know, there are no specific studies on the relationship between EDS, depressive symptoms, and SI among Chinese college students. This study found that (1) both EDS and depressive symptoms were significantly associated with SI in the Chinese college student population, and (2) depressive symptoms partially mediated the relationship between EDS and SI.

In the study, the prevalence of EDS among college students was approximately 52.8%, which was higher than most previous studies [21,26,32,33], and consistent with our study hypothesis. Possible causes include COVID-19 stressors [34], poor economic status, mental health issues such as anxiety [35,36], and lockdown [25]. The prevalence of depressive symptoms among college students was 25.5%, which is essentially the same as reported in a previous review of the prevalence of depressive symptoms during the COVD-19 pandemic [11]. We should strengthen the screening and assessment of depression among college students, expand social support, develop appropriate depression prevention programs for high-risk groups, and provide timely psychiatric assessment and treatment for depressed college students. This study found that the proportion of college students who had SI in the past year was approximately 20.1%. The prevalence of SI among Chinese college students presented in past studies ranged from 9.1% to 26.2% [37]. This suggests that the prevalence of SI among Chinese college students remains at a high level.

Many studies have demonstrated a strong correlation between EDS and depressive symptoms in different age groups [21,38,39,40], which is consistent with the findings of the present study on a population of Chinese college students. In cohort studies of adolescents and children, EDS is a predictor of one-year depressive symptoms [40]. In addition, one study has shown a stronger correlation between EDS and depressive symptoms in adults younger than 30 years of age [41]. A longitudinal study of sleep disturbance and depression during the COVID-19 pandemic also illustrated that sleep disturbance predicted the development and persistence of depression [42]. Current research suggests that the possible causes of the association between EDS and depressive symptoms are alterations in the homeostatic regulation and circadian regulation of physiological pathways [43] or abnormalities in the neuroendocrine system [44]. Overall, the mechanisms underlying the association between EDS and depressive symptoms are complex and require further study [45].

In the study of correlates of SI, both EDS and depressive symptoms were correlated, and the correlation between EDS and SI was more significant in females. Excessive daytime sleepiness is associated with SI in depressed children or adolescents [46]. In addition, in the adolescent population, EDS was significantly associated with suicidal ideation during a one-year follow-up period. In adults attending a hospital or psychiatric hospital, EDS was associated with SI during a one-month follow-up period [18]. This study demonstrated a correlation between EDS and SI in a college student population. The relationship between SI and depressive symptoms in various age groups, including college students, has been demonstrated, i.e., depressive symptoms as an independent risk factor for SI [47,48,49,50]. This was further confirmed by the present study. As for the gender difference, it may be due to the gender difference in EDS, which is similar to the results of previous studies [11,26,51]. A study noted that female medical students had more difficulty managing time balance and were more likely to have EDS, making them more likely to have internal insecurities and feelings of loss, which in turn led to a lower quality of life [51]. It has also been noted that women with EDS are more likely to have lower self-assessed health scores than men in the college student population [33]. This may account for the stronger correlation between EDS and SI in female college students.

The results of the mediation study suggested that the association between EDS and SI in college students was mediated to some extent by depressive symptoms. Previous research has suggested that individuals with EDS suffer from more severe depression, which may influence the risk of suicidal ideation [26,52]. The study confirmed the relationship between the three. This suggests additional ideas for crisis intervention for college students. We can consider the EDS college student population as a high-risk SI population. In order to avoid the occurrence of SI in college students, we can develop appropriate strategies. First, we can take measures for college students with EDS, which can be done through medication or other treatments to ensure quality sleep at night for college students to avoid EDS. In addition, we should strengthen the monitoring and assessment of depressive symptoms in college students with EDS, and conduct psychological interviews or offer medication if necessary to avoid SI caused by depressive symptoms.

There are several limitations of this study. First, this study is a cross-sectional study and cannot determine the causal relationship between variables, which needs to be verified by further longitudinal studies. In addition, this assessment was based on a self-report questionnaire rather than a diagnostic interview, which may be subject to information bias. Furthermore, we did not include variables associated with the COVID-19 pandemic to explore the influence of these variables on our models. Finally, there were some differences in mediating effects between males and females, and further research is needed to validate the relationship between EDS and SI.

## 5. Conclusions

The study of a sample of Chinese college students showed that both EDS and depressive symptoms were correlated with SI in college students. Furthermore, the association between EDS and SI was partially mediated by depressive symptoms. For college students with EDS, the emphasis should be on the prevention of depressive episodes and thus the occurrence of suicide-related events. In addition, the neurophysiological mechanisms between EDS, depressive symptoms, and SI in college students need to be further explored.

## Figures and Tables

**Figure 1 ijerph-19-16334-f001:**
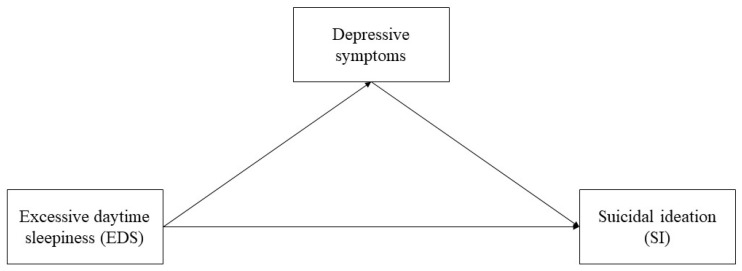
The mediating effect model of depressive symptoms on the relationship between excessive daytime sleepiness and suicidal ideation.

**Table 1 ijerph-19-16334-t001:** Demographics and clinical characteristics between participants with and without suicidal ideation (*n* = 6944).

Variables	SI *n* = 1394	Without SI *n* = 5550	χ^2^/Z	*p* Value	OR (95% CI)
Age (years), median (IQR)	19.00 (18.00–20.00)	19.00 (19.00–20.00)	−0.99	0.320	0.99 (0.94–1.03)
Gender, *n* (%)			9.86	0.002	
Male	473 (33.9)	2136 (38.5)			Ref.
Female	921 (66.1)	3414 (61.5)			1.22 (1.08–1.38)
BMI (kg/m^2^), median (IQR)	20.55 (18.90–22.86)	20.55 (18.94–22.77)	−0.02	0.983	1.00 (0.98–1.02)
Community, *n* (%)			29.29	<0.001	
Urban	677 (48.6)	2251 (40.6)			Ref.
Rural	717 (51.4)	3299 (59.4)			0.72 (0.64–0.81)
Nationality, *n* (%)			0.32	0.571	
Han	1306 (93.7)	5222 (94.1)			Ref.
Others	88 (6.3)	328 (5.9)			0.93 (0.73–1.19)
Grade, *n* (%)			24.51	<0.001	
Freshman	518 (37.2)	1780 (32.1)			Ref.
Sophomore	460 (33.0)	2195 (39.5)			0.99 (0.81–1.21)
Junior	247 (17.7)	1000 (18.0)			0.71 (0.58–0.87)
Senior and others	169 (12.1)	575 (10.4)			0.84 (0.67–1.05)
Only child, *n* (%)	611 (43.8)	2220 (40.0)	6.77	0.009	1.17 (1.04–1.32)
Major, *n* (%)			35.67	<0.001	
Others	1049 (75.2)	4567 (82.3)			Ref.
Medicine	345 (24.8)	983 (17.7)			1.53 (1.33–1.76)
Smoking, *n* (%)	100 (7.2)	240 (4.3)	19.42	<0.001	1.71 (1.34–2.18)
Drinking, *n* (%)	688 (49.4)	2167 (39.0)	48.91	<0.001	1.52 (1.35–1.71)
Academic performance, *n* (%)			124.05	<0.001	
Poor	206 (14.8)	358 (6.5)			Ref.
Fair	883 (63.3)	3498 (63.0)			3.12 (2.59–3.94)
Good	305 (21.9)	1694 (30.5)			1.40 (1.22–1.62)
Economic status, *n* (%)			19.46	<0.001	
Poor	250 (18.0)	739 (13.3)			Ref.
Fair	979 (70.2)	4119 (74.2)			1.42 (1.14–1.77)
Good	165 (11.8)	692 (12.5)			1.00 (0.83–1.20)
Not single, *n* (%)	357 (25.6)	1433 (25.8)	0.03	0.873	0.99 (0.87–1.13)
School activity, *n* (%)			19.82	<0.001	
Attend	1259 (90.3)	5201 (93.7)			Ref.
Never attend	135 (9.7)	349 (6.3)			1.60 (1.30–1.97)
Game time (h/d), median (IQR)	1.00 (0.00–2.00)	1.00 (0.00–2.00)	−4.76	<0.001	1.11 (1.07–1.15)
Media time (h/d), median (IQR)	3.00 (2.00–4.00)	3.00 (2.00–4.00)	−6.19	<0.001	1.07 (1.05–1.10)
Sports time (h/w), median (IQR)	2.00 (1.00–4.00)	2.00 (1.00–4.00)	−6.93	<0.001	0.94 (0.92–0.96)
Sleep quality, *n* (%)			214.02	<0.001	
Poor	200 (14.3)	327 (5.9)			Ref.
General	644 (46.2)	1968 (35.5)			3.62 (2.97–4.41)
Good	550 (39.5)	3255 (58.6)			1.94 (1.71–2.20)
EDS, *n* (%)	864 (62.0)	2806 (50.6)	58.33	<0.001	1.59 (1.41–1.80)
Depressive symptoms, *n* (%)	764 (54.8)	1007 (18.1)	788.21	<0.001	5.47 (4.83–6.20)

SI: Suicidal ideation; BMI: Body mass index; EDS: Excessive daytime sleepiness.

**Table 2 ijerph-19-16334-t002:** Associations between excessive daytime sleepiness, depressive symptoms, and suicidal ideation.

Variables	Model 1				Model 2			
	β	OR	95% CI	*p*	β	OR	95% CI	*p*
Total								
EDS	0.24	1.27	1.12–1.44	<0.001	0.14	1.15	1.01–1.32	0.035
Depressive symptoms	1.66	5.27	4.64–5.99	<0.001	1.47	4.36	3.80–5.00	<0.001
Male								
EDS	0.16	1.17	0.95–1.45	0.138	0.09	1.10	0.88–1.36	0.402
Depressive symptoms	1.50	4.48	3.63–5.54	<0.001	1.32	3.74	2.98–4.69	<0.001
Female								
EDS	0.23	1.26	1.07–1.48	0.007	0.19	1.21	1.02–1.43	0.027
Depressive symptoms	1.82	6.16	5.24–7.24	<0.001	1.55	4.70	3.95–5.60	<0.001

Model 1: Univariate logistic regression; Model 2: Multivariate logistic regression, controlling for gender, community, grade, only child, major, smoking, drinking, academic status, economic status, school activity, game time, media time, sport time, and sleep quality in Table 1.

**Table 3 ijerph-19-16334-t003:** The mediating effect of depressive symptoms on the relationship between excessive daytime sleepiness and suicidal ideation.

Paths	Effect	95% CI	SE	z	*p*	Effect Size (%)
Total						50.41
Total effect	0.29	0.16–0.42	0.07	4.29	<0.001	
Direct effect	0.14	0.01–0.28	0.07	2.11	0.035	
Indirect effect	0.15	0.11–0.18	0.02	9.05	<0.001	
Male						58.33
Total effect	0.24	0.03–0.46	0.11	2.20	0.027	
Direct effect	0.10	−0.12–0.32	0.11	0.89	0.374	
Indirect effect	0.14	0.09–0.19	0.03	5.65	<0.001	
Female						42.98
Total effect	0.33	0.16–0.50	0.09	3.87	<0.001	
Direct effect	0.19	0.02–0.36	0.09	2.20	0.028	
Indirect effect	0.14	0.10–0.18	0.02	7.02	<0.001	

Total effect = direct effect + indirect effect; Direct effect: EDS—SI; Indirect effect: EDS—depressive symptoms—SI; Total: controlling for gender, community, grade, only child, major, smoking, drinking, academic status, economic status, school activity, game time, media time, sport time, and sleep quality; Male and female: controlling for community, grade, only child, major, smoking, drinking, academic status, economic status, school activity, game time, media time, sport time, and sleep quality.

## Data Availability

The data are not publicly available due to privacy and research ethical restrictions.

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
