# Peer review of "Depressive Symptoms as a Mediator between Excessive Daytime Sleepiness and Suicidal Ideation among Chinese College Students"

_ijerph, 2022, doi:10.3390/ijerph192316334_

Round 1

Reviewer 1 Report

The work presented by the authors is well-written and appealing to readers. It appears pretty clear and fluent. The interest in the topic is high, but the completeness and pleasantness of the work are lost on a doubt that is important in terms of methodology that I hope the authors can resolve. The study was conducted during a globally recognized pandemic period. The authors do not mention possible links between what they observe in this specific window and pandemic outcomes regarding mental health. This aspect, even if not observed at the methodological level, should still be discussed as a limitation in the conclusions.

 I would find it useful to include the figure of mediation in the results.

Author Response

Point 1: The work presented by the authors is well-written and appealing to readers. It appears pretty clear and fluent. The interest in the topic is high, but the completeness and pleasantness of the work are lost on a doubt that is important in terms of methodology that I hope the authors can resolve. The study was conducted during a globally recognized pandemic period. The authors do not mention possible links between what they observe in this specific window and pandemic outcomes regarding mental health. This aspect, even if not observed at the methodological level, should still be discussed as a limitation in the conclusions.

Response: We appreciate the comments. This study was conducted during the COVID-19 pandemic, and many variables were affected by the pandemic. In the background section, we added the current dire situation of the pandemic, and in the discussion section, we compared and explored the study variables. The main findings are as follows: (i) the prevalence of EDS among college students during the pandemic was higher than in most previous studies; (ii) depressive symptoms and SI were largely consistent with other studies during the pandemic; and (iii) it was confirmed that sleep disturbance among college students predicted the onset and progression of depression in previous studies. In addition, the investigation did not include some variables related to COVID-19 as in the Foster and Rankin study, which has been added to the article as a limitation.

Marked:

Lines 38-42, Page 1. Currently in the midst of the COVID-19 pandemic, which adversely affected the men-tal health of both people with mental illness and healthy populations and was directly related to multiple suicide triggers, such as stress, unemployment, and alcohol use, the situation was extremely negative for suicide prevention now.

Lines 72-75, Page 2. Currently during the COVID-19 pneumonia pandemic, sleep indicators among college students have deteriorated significantly, including sleep quality, sleep latency, sleep duration, sleep disturbance, and increased use of sleep medication. Therefore, exces-sive daytime sleepiness among college students may be more severe.

Lines 221-223, Page 7. In the study, the prevalence of EDS among college students was approximately 52.8%, which was higher than most previous studies, and consistent with our study hypothesis. Possible causes include COVID-19 stressors, poor economic status, mental health issues such as anxiety, and lockdown.

Lines 238-240, Page 7. A longitudinal study of sleep disturbance and depression during the COVID-19 pan-demic also illustrated that sleep disturbance predicted the development and persis-tence of depression.

Lines 279-280, Page 8. Furthermore, we did not include variables associated with the COVID-19 pandemic to explore the influence of these variables on our models.

Point 2: I would find it useful to include the figure of mediation in the results.

Response: Thank you very much for your suggestion. The complete figures for the mediating effects model have been included in the results section.

Reviewer 2 Report

My first impressions that this was a well designed and for the most part well-reported study.  However as I carried on reading I felt there were significant flaws/gaps that need to be addressed.  I suggest that further statistical expertise is sought.

a) There is a recent paper (Open access) that you will find very helpful in both the introduction and discussion.

Self-Reported Sleep during the COVID Lockdown in a Sample of UK University Students and Staff

Foster and Rankin 2022

b) There is a confusion at times between EDS and ESS- please use one or the other.

c) In table 3 I am unclear where/how total/direct/indirect effect are defined and what they relate to.

d) This is the main flaw.  There appears to be minimal connection between table 1 and table 3.  There may be a number of variables in your study (e.g drinking - but how is this defined) that could contribute to if not explain some of your findings (again see Foster and Rankin).  I suggest you re analyse your data to take account of the co-variates in table 1.  Your discussion should then consider these results

e). The study was conducted during the Covid pandemic/lockdown.  There is no recognition of this in your discussion.  There is a literature concerning the impact of lockdown/sleep/health and well-being etc on student populations.  Foster and Rankin will provide some indications of papers for you to consider.

f). Please state which Ethics Committee or other approval body approved and monitored the study.

g). The discussion will need a significant re-write following the re-analysis

Author Response

Thank you very much for your review of our article. We truly appreciate the invaluable comments. Our specific responses, point-to-point to the comments, are described as follows:

a) There is a recent paper (Open access) that you will find very helpful in both the introduction and discussion.

Self-Reported Sleep during the COVID Lockdown in a Sample of UK University Students and Staff (Foster and Rankin 2022)

Response: Thank you very much for your suggestion. I have read this article carefully and found it to be of great importance to my research and has given me some inspiration for future research. It has now been used as a reference for background and discussion, and your recommendation is much appreciated.

b) There is a confusion at times between EDS and ESS please use one or the other.

Response: Thank you for your comments. Excessive daytime sleepiness (EDS) was assessed by the Epworth Sleepiness Scale (ESS). ESS is described in the article only in the method. We have removed ESS from the article.

c) In table 3 I am unclear where/how total/direct/indirect effect are defined and what they relate to.

Response: We appreciate the comments. We have added the relevant content to the statistical analysis and to the notes section of Table 3.

Marked:

Lines 146-148, Page 4. The mediating effect model was described by total, direct and indirect effects. The path of the direct effect was EDS - SI, the path of the indirect effect was EDS - depressive symptoms - SI, and the total effect was the sum of the two.

Lines 208-209, Page 6. Total effect = direct effect + indirect effect; Direct effect: EDS – SI; Indirect effect: EDS – depressive symptoms – SI.

d) This is the main flaw.  There appears to be minimal connection between table 1 and table 3.  There may be a number of variables in your study (e.g drinking - but how is this defined) that could contribute to if not explain some of your findings (again see Foster and Rankin).  I suggest you re analyse your data to take account of the co-variates in table 1.  Your discussion should then consider these results

Response: Please allow me to apologize for my mistake. In this study, all mediating effect models were performed controlling for the meaningful covariates in Figure 1. Due to our oversight, this was not explained clearly. The content about covariates has now been added to the notes section of Table 3.

Marked: Lines 209-213, Page 4. Total: controlling for gender, community, grade, single child, major, smoking, drinking, academic status, economic status, school activity, game time, media time, sport time and sleep quality; Male and female: controlling for community, grade, single child, major, smoking, drinking, academic status, economic status, school activity, game time, media time, sport time and sleep quality.

e). The study was conducted during the Covid pandemic/lockdown.  There is no recognition of this in your discussion.  There is a literature concerning the impact of lockdown/sleep/health and well-being etc on student populations.  Foster and Rankin will provide some indications of papers for you to consider.

Response: Thank you very much for your suggestion. This study was conducted during the COVID-19 pandemic, and many variables were affected by the pandemic. In the background section, we added the current dire situation of the pandemic, and in the discussion section, we compared and explored the study variables. The main findings are as follows: (i) the prevalence of EDS among college students during the pandemic was higher than in most previous studies; (ii) depressive symptoms and SI were largely consistent with other studies during the pandemic; and (iii) it was confirmed that sleep disturbance among college students predicted the onset and progression of depression in previous studies. In addition, the investigation did not include some variables related to COVID-19 as in the Foster and Rankin study, which has been added to the article as a limitation.

Marked:

Lines 38-42, Page 1. Currently in the midst of the COVID-19 pandemic, which adversely affected the men-tal health of both people with mental illness and healthy populations and was directly related to multiple suicide triggers, such as stress, unemployment, and alcohol use, the situation was extremely negative for suicide prevention now.

Lines 72-75, Page 2. Currently during the COVID-19 pneumonia pandemic, sleep indicators among college students have deteriorated significantly, including sleep quality, sleep latency, sleep duration, sleep disturbance, and increased use of sleep medication. Therefore, exces-sive daytime sleepiness among college students may be more severe.

Lines 221-223, Page 7. In the study, the prevalence of EDS among college students was approximately 52.8%, which was higher than most previous studies, and consistent with our study hypothesis. Possible causes include COVID-19 stressors, poor economic status, mental health issues such as anxiety, and lockdown.

Lines 238-240, Page 7. A longitudinal study of sleep disturbance and depression during the COVID-19 pandemic also illustrated that sleep disturbance predicted the development and persistence of depression.

Lines 279-280, Page 8. Furthermore, we did not include variables associated with the COVID-19 pandemic to explore the influence of these variables on our models.

f). Please state which Ethics Committee or other approval body approved and monitored the study.

Response: Thank you for your comments, and we have added the approval number as suggested.

Marked: Lines 297-299, Page 8. The study was conducted in accordance with the Decla-ration of Helsinki, and approved by the research ethics boards of School of Public Health, Shandong University (2018-01-18, #20180123).

g). The discussion will need a significant re-write following the re-analysis

Response: Thank you for your comments. We have made some minor changes to the discussion section. If you have any further suggestions, we are always ready to listen to them.

Round 2

Reviewer 2 Report

Thank you for your careful responses to my comments and the clear way you responded both in your reply letter.  In future ensure that you clearly highlight changes throughout the manuscript.  I had to search for the Foster and Rankin reference which I think is a key recent work in this area.